# Residual Interactions of LL-37 with POPC and POPE:POPG Bilayer Model Studied by All-Atom Molecular Dynamics Simulation

**DOI:** 10.3390/ijms232113413

**Published:** 2022-11-02

**Authors:** Muhammad Yusuf, Wanda Destiarani, Ade Rizqi Ridwan Firdaus, Fauzian Giansyah Rohmatulloh, Mia Tria Novianti, Gita Widya Pradini, Reiva Farah Dwiyana

**Affiliations:** 1Department of Chemistry, Faculty of Mathematics and Natural Sciences, Universitas Padjadjaran, Bandung 45363, Indonesia; 2Research Center for Molecular Biotechnology and Bioinformatics, Universitas Padjadjaran, Bandung 45363, Indonesia; 3Department of Microbiology and Parasitology, Faculty of Medicine, Universitas Padjadjaran, Bandung 45363, Indonesia; 4Department of Dermatology and Venereology, Faculty of Medicine, Universitas Padjadjaran, Bandung 45363, Indonesia

**Keywords:** LL-37, molecular dynamics simulation, POPC, POPE:POPG, residual interactions

## Abstract

LL-37 is a membrane-active antimicrobial peptide (AMP) that could disrupt the integrity of bacterial membranes due to its inherent cationic and amphipathic nature. Developing a shorter derivative of a long peptide such as LL-37 is of great interest, as it can reduce production costs and cytotoxicity. However, more detailed information about the residual interaction between LL-37 and the membrane is required for further optimization. Previously, molecular dynamics simulation using mixed all-atom and united-atom force fields showed that LL-37 could penetrate the bilayer membrane. This study aimed to perform all-atom molecular dynamics simulations, highlighting the residual interaction of LL-37 with the simplest model of the bacterial membrane, POPE:POPG (2:1), and compare its interaction with the POPC, which represents the eukaryotic membrane. The result showed leucine–leucine as the leading residues of LL-37 that first contact the membrane surface. Then, the cationic peptide of LL-37 started to penetrate the membrane by developing salt bridges between positively charged amino acids, Lys–Arg, and the exposed phosphate group of POPE:POPG, which is shielded in POPC. Residues 18 to 29 are suggested as the core region of LL-37, as they actively interact with the POPE:POPG membrane, not POPC. These results could provide a basis for modifying the amino acid sequence of LL-37 and developing a more efficient design for LL-37 derivatives.

## 1. Introduction

Membrane-active antimicrobial peptides (AMPs) are found in numerous organisms and have significant potential as drugs against progressively multiresistant pathogens [1]. The majority of these peptides eliminate microorganisms by physically interacting with and disrupting their cell membranes [2]. Cathelicidins are one of the most popular and well-studied classes of AMPs. The only class naturally expressed in humans is LL-37, which is derived by cleavage of the precursor human cationic antimicrobial protein 18 (hCAP18). It also appears to have a function as an immunomodulator, in addition to its role as an antibacterial peptide [3]. LL-37 is similar to that described for defensins and works synergistically in the immune system as an agent against bacteria, some fungi, parasites, and specific viruses [4,5,6]. The direct antimicrobial activity of LL-37 is considerably mediated by the disruption of the integrity of bacterial membranes due to its amphipathic and cationic nature [7,8]. However, producing a long peptide such as LL-37 is costly. Hence, shorter AMPs are attracting more interest. For example, KR-12 (KRIVQRIKDFLR; 18–29) is the shortest peptide with antibacterial activity derived from LL-37. This peptide retains the core region with an amphipathic helix structure from LL-37 and consists of five cationic residues [9,10,11]. Although the MIC value of LL-37 (5 µM) is lower than that of KR-12 (40 µM) [12,13], as a short peptide, KR-12 has both antimicrobial and antiendotoxic activities, without mammalian cytotoxicity [10]. The differences in the length and structure of both peptides might affect their activity as well as their mechanisms of action as AMPs. The exact antimicrobial molecular mechanisms of LL-37 are not completely understood, although many models have been proposed in recent years [14,15,16]. 

The two major lipid components that compose bacterial cell membranes are phosphatidylethanolamine (PE) and phosphatidylglycerol (PG). In fact, the inner membrane of *Escherichia coli* contains 70–80% PE, 20–25% PG, and 5% cardiolipin [17]. In this study, the POPE:POPG (2:1) membrane was modeled to resemble the fundamental character of most Gram-negative bacterial membranes. PG lipids, in particular, exert a net negative charge on bacterial membranes [18]. The phosphatidylcholine (POPC) membrane can represent the human cell membrane since it is extensively used as a mimic for eukaryotic membranes. It is also the most common phospholipid type [19].

Molecular studies at the atomic level to observe the molecular mechanisms and interactions that occur can be carried out with molecular dynamics simulations (MD). This method can reveal the interactions of a system at a spatiotemporal resolution, which cannot be accomplished with any other method [20,21]. Although a previous in silico study already showed that LL-37 could penetrate the POPG bilayer membrane [22], the residual interactions that facilitate how it works are yet to be explained. In addition, the previous simulations were conducted using a mix of united- and all-atom force fields [22]. Thus, here, we conducted more extensive all-atom MD simulations of LL-37 embedded in a lipid bilayer membrane and a water solvent box, including a more realistic POPE:POPG model to represent the bacterial membrane. This study aimed to investigate the residual interaction of peptide–membrane as part of the mechanism of LL-37 penetration with POPE:POPG (1-palmitoyl-2-oleoyl phosphatidylethanolamine/glycerol) and POPC (1-palmitoyl-2-oleoyl-sn-glycero-3-phosphocholine) as simple representations of bacterial and human cell membranes, respectively. These results could provide a basis for modifying the amino acid sequence of LL-37 and developing a more efficient design for LL-37 derivatives.

## 2. Results

### 2.1. The LL-37–POPC/POPE:POPG Membrane Systems

The calculation of the average area per lipid is useful in bilayer remodeling and membrane molecular dynamics (MD) simulations. The extent to which osmotic pressure deforms lipid bilayers is also critical to understanding how lipid–peptide interactions affect membrane functions. The results in Figure 1 show that, consistent with the liquid-disordered phase, the POPC membrane simulation exhibited an average area per lipid of 68.9 Å^2^, while the value for POPE:POPG was 60.4 Å^2^. These results agree with the previous study of membrane MD simulations with the same lipid composition [17,23].

### 2.2. Interactions of LL-37 with POPC and POPE:POPG Membrane

The 600-ns-long simulations showed the attachment of LL-37 to the POPE:POPG membrane caused by the affinity of rich, positively charged LL-37 and the negatively charged surface of the POPE:POPG membrane. During the whole simulation, the LL-37 in the POPC system membrane showed several changes to obtain the fittest conformation to interact with the membrane surface (Figure 2a). On the other hand, LL-37, particularly residues 18–29, persistently attached to the POPE:POPG membrane while retaining its helical conformations (Figure 2b). The simulated trajectory analysis found that the guanidine group of arginine played a more significant role in its interaction with the POPE:POPG membrane than lysine, although arginine (R) and lysine (K) were both positively charged. Furthermore, by evaluating the solvent accessibility of membrane surfaces using a molecular probe with a radius of 1.4 Å, we could explain the possible reason behind the selectivity of antimicrobial peptides between mammalian and bacterial membranes. The trimethyl group of POPC’s choline shielded the positivity of its nitrogen, thus rendering the phosphate group of POPC inaccessible to the solvent (Figure 2c). Therefore, LL-37 could not attach to the POPC membrane surface. On the other hand, if only considering the 2D structure of POPE, it has a positively charged ethanolamine that is supposed to repel LL-37. Interestingly, our simulation showed that the amine group of POPE formed intramolecular electrostatic interactions with its phosphate group. For this reason, the POPE negatively charged phosphate group is more exposed to the surface, thus attracting positive residues in LL-37 (Figure 2d). The atomic-level detail of these interactions is represented better in our all-atom MD simulations than in the united-atom and coarse-grained models. The number of interaction sites representing a molecule is one of the major differences between force fields. Each atom in an all-atom model corresponds to an interaction site, while, in united-atom models, an interaction site might represent a collection of atoms. It is similar to a coarse-grained method, in which some variables of the molecule are forfeited [24]. 

The residual interactions of the N-terminal of LL-37 with the POPC membrane surface mostly consist of hydrophobic interactions involving Leu and Phe. Several electrostatic interactions are also facilitated by the charged amino acids, such as Asp, Glu, Lys, and Arg (Figure 3a), while the remaining residues of LL-37 did not come into contact with the membrane surface during the conformational changes throughout the simulations. In contrast, LL-37 residues had relatively stable full contact with the surface of the POPE:POPG membrane. It was facilitated by the electrostatic interactions of positively charged base groups such as arginine and lysine on LL-37 with negatively charged phosphate and hydroxyl groups on the POPE:POPG membrane (Figure 3b). In total, there were 16 and 33 residual hydrophobic and electrostatic interactions, respectively, in the POPC system (Appendix A). Meanwhile, the residual interactions of LL-37 with POPE:POPG are higher than with POPC, with 41 and 48 hydrophobic and electrostatic interactions (Appendix A).

The interactions of LL-37 with the POPC and POPE:POPG membranes were determined quantitatively using the interaction energy. The calculation of the binding free energy (ΔG°) between LL-37 and each of its membrane models was performed using the MM/PBSA method at 60 ns intervals during the 600-ns-length molecular dynamics simulations (Table 1). The results showed that LL-37 had a higher affinity towards the POPE:POPG membrane, with −85.4955 kcal/mol, while this value was −19.3089 for the POPC membrane. These findings are in agreement with the previous analysis of the interaction energy. 

Residues that play a significant role in the LL-37–POPE:POPG membrane interaction were identified (Figure 4). These residues were Leu1, Arg7, Lys8, Lys10, Lys12, Lys15, Lys18, Arg19, Arg23, Lys25, Arg29, and Arg34, which mostly consist of basic amino acids. Lys18, Arg23, Lys25, and Arg29 in the core region suggested that the antimicrobial activity of AMPs also depends on the constituents of positively charged amino acids. The residues Asp4, Glu11, Glu16, and Asp26 have positive interaction energy, which can be interpreted as a repulsive interaction with the negatively charged membrane surface of POPE:POPG because of their acidic properties. However, these positive interaction energies are compensated by the majority of negative energies from the other residues. As for the POPC membrane, the interaction only occurred in the N-terminal of LL-37, while the core region (18–29 residues) had almost zero interaction energy as it was far from the membrane surface. The core region could not attach strongly to the membrane surface because of the weak van der Waals interaction with the trimethyl groups of POPC. 

The helicity percentage (%) was used to monitor the conformational changes of LL-37 during interaction with the POPC and POPE:POPG membranes. LL-37 retains its helical conformation while attached to the POPE:POPG membrane surface. However, in the POPC membrane system, the first 16 residues of LL-37 have low helicity (%), caused by its mobility to find the most stable conformation to interact with the membrane surface (Figure 5).

### 2.3. Alteration of POPC and POPG Membranes

The interaction with LL-37 also impacts the convergence of the POPG and POPC membranes themselves. To observe the dynamics of LL-37 in the membrane, electron density mapping was performed. The distribution of the mass density of LL-37 and the membranes (Figure 6) was represented by the electron density profile as a function of distance from the membrane center (the bilayer center is at z = 0). Based on the comparison of the membrane and the LL-37 density distribution at initial configurations and after 600-ns-long simulation trajectories, it can be concluded that LL-37 starts to penetrate the membrane (Figure 6a,b) or avoid the membrane instead (Figure 6c,d).

Distance analysis using the cpptraj program in Amber20 was carried out to show the average distance between the center of mass of LL-37 and the center of the POPE:POPG and POPC bilayer membranes (Figure 6e). During 600 ns simulations, there is a gap in the distance between the two, which is reciprocally linear to the stability of the interaction. It can be seen that LL-37 and the center of the POPE:POPG bilayer membrane have a lower average distance of ~42 Å. In the POPC system, the distance fluctuates relatively more and is greater than that of POPE:POPG, which is around ~50 Å. The closer the average distance, the more stable the penetration of the LL-37 peptide to the membrane. 

The solvent-accessible surface area (SASA) of the LL-37 peptide is described as the peptide region sufficiently exposed to interact with adjacent solvent molecules. SASA can be presented as a factor that contributes to protein stability and folding studies, and it is defined as the hypothetical center of the solvent sphere involving van der Waals’s contact with the molecule’s surface [25,26]. In the whole 600 ns simulation, the SASA values of LL-37 in POPE:POPG are lower than in the POPC membrane. It shows that as LL-37 is attached to the POPE:POPG membrane, it becomes less accessible to the solvent (Figure 6f).

## 3. Discussion

Antimicrobial peptide LL-37 was simulated in the presence of bacterial cell membrane mimics, which were POPE:POPG (2:1) membranes. In terms of structure, LL-37 is a linear peptide that is rich in basic residues (5 Arg, 6 Lys), resulting in a +6 net charge at physiological pH. Because approximately 35% of the residues are hydrophobic, it adopts an amphipathic helical shape with a small but well-defined hydrophobic sector. All of these properties are required for oligomerization in aqueous salt solutions, as well as peptide–membrane interactions [14], because their cationic and amphipathic characteristics mimic the antimicrobial peptides that prokaryotes excrete to kill other bacteria [27].

Preliminary analysis as an assessment of the peptide–membrane systems was conducted via area per lipid calculation (Figure 1). It showed that both systems have similar values to the average area per lipid for the POPC and POPE:POPG lipid bilayers in other studies [17,23]. Thus, it can be said that the peptide–membrane systems in our simulation were confirmed to be rational. 

At the atomic level, the first contact between LL-37 and the membrane is started with leucine residues at the N-terminal, as the most hydrophobic amino acid (Figure 3a,b). In a previous study, deleting these two leucine residues did not affect the antibacterial activity but reduced the cytotoxicity [28]. In the POPC membrane system, a few electrostatic interactions were facilitated by the charged amino acids, such as Asp, Glu, Lys, and Arg (Figure 3a). At the same time, the remaining residues of LL-37 did not come into contact with the POPC membrane surface. In contrast, LL-37 residues had relatively stable full contact with the surface of the POPE:POPG membrane. This is also facilitated by the electrostatic interactions of arginine and lysine on LL-37 with the negatively charged phosphate and hydroxyl groups on the POPE:POPG membrane (Figure 3b). In total, the number of residual hydrophobic and electrostatic interactions in the POPE:POPG system is two-fold higher than that in the POPC system (Appendix A). As for the general observation, it was shown that LL-37 was fully buried in the surface of the POPE:POPG membrane (Figure 2b), contrasting the surface of the POPC membrane, where it was barely in contact, and only the N-terminal part had a hydrophobic interaction with the POPC membrane (Figure 2a).

The basic amino acid, which is positively charged, also plays a significant role in LL-37’s penetration in the membrane process. The Lys and Arg residues develop emerging salt bridges with the headgroup of POPE:POPG and the exposed phosphate group, resulting in strong electrostatic interactions, and hydrophobic interactions also contribute to LL-37–membrane binding. Our simulation found that the ethanolamine group bends towards the phosphate group, thus exposing the negatively charged phosphate group in the POPE:POPG membrane system (Figure 2d). Meanwhile, the trimethyl of the choline group covered the phosphate group of POPC, thus making the surface less susceptible to the cationic peptide (Figure 2c). We suggested this phenomenon as the source of the positively charged peptide activity on POPE, but not POPC. To the best of our knowledge, this suggestion is, for the first time, proposed as the cationic peptide selectivity determinant in bacterial (POPE:POPG) membranes over mammalian (POPC) membranes. The selectivity of LL-37 as an AMP was also observed in this study, consistent with earlier reports [29,30]. This is in accordance with the binding free energy results showing that the affinity of LL-37 towards the POPE:POPG membrane was around −85 kcal/mol, which is four times higher than the POPC membrane, at around −19 kcal/mol (Table 1).

This study also revealed the residues that play an essential role in the interaction of the LL-37–POPE:POPG membrane, which mainly consist of Lys and Arg, whereas amino acids with specific properties, such as basic (arginine or lysine) or hydrophobic amino acids (phenylalanine or tryptophan, leucine, and alanine), are usually used. Specifically, Lys18, Arg23, Lys25, and Arg29 in the core region suggested that the antimicrobial activity of AMPs also depends on the constituents of positively charged amino acids (Figure 4). Furthermore, a similar finding was experimentally proven by alanine scanning, where positively charged residues between positions 18 and 29 were changed to alanine, resulting in a substantial decrease in antimicrobial activity [31,32]. 

During the 600-ns-long simulations, LL-37 retained its helical properties, particularly in the core region, which consists of residues 17–29 (Figure 5). However, for this region, it was experimentally proven that it remained toxic to human cells [33]. The study from Jacob et al. showed that the removal of Phe17 to form a peptide with residues 18–29 still results in antimicrobial activity, with no toxicity to mammalian cells [10]. These findings demonstrate that the loss of one hydrophobic phenylalanine can drastically decrease toxicity, which is in line with our results. In this simulation, residues 18–29 were observed to be the crucial region of LL-37, as they actively interact with the POPE:POPG membrane while retaining its helical properties. These are also in agreement with the previous study of LL-37 as a helix rich in positively charged side chains, allowing efficient interaction with anionic phosphatidylglycerols in bacterial membranes [34,35]. For some other AMPs, helical formation is also an important structural element that possesses antimicrobial capability [22]. 

Evidence of LL-37 and POPE:POPG membrane interactions can also be identified as membrane property alterations. The electron density profile of LL-37 and the POPE:POPG membranes changed after the 600 ns simulation, showing that the electron density peak of LL-37 shifted closer to the center of the membrane, while, in the POPC membrane, LL-37 shifted further away, indicating the repulsion of the membrane surface (Figure 6b–d). The shifting of the AMP’s electron density toward the center of the membrane can be presented as the penetration process in the interaction study of AMPs and the membrane surface [36]. These findings are consistent with previous results that also showed the changes in AMPs’ electron density; it becomes closer to the center of the membrane as its contact with the membrane surface increases [22].

Moreover, the distance between LL-37 and the center of the POPC bilayer membrane is further than that in POPE:POPG (Figure 6e). The distance parameter is seldom used in this field of study, but it can explain the common visualized distance in many studies between the AMP and membrane quantitatively, as its values will decline the closer to the membrane it is, and they will rise the farther away it is. The result is also consistent with the SASA results, as they showed that LL-37 in the POPE:POPG membrane is less accessible because of the attachment to the membrane surface (Figure 6f). This can be interpreted to lead to the idea of how the interactions between AMPs and membranes work [37]. The deeper LL-37 is buried within the hydrophobic environment of the POPE:POPG membrane surface, the less accessible it becomes for the solvent.

Our observation on the mechanism of LL-37 in penetrating the POPE:POPG membrane is similar to previous studies of LL-37 that proposed a non-pore carpet-like mechanism that showed an α-helical structure with the axis lying in the plane of the membrane [15,22,38]. Other studies proposed that cationic AMPs enter cells via an energy-dependent endocytic mechanism known as micropinocytosis [39,40]. Another experiment with D-enantiomers of LL-37, along with fluorescence assays, confirmed the capacity of the peptide to attach to the membrane [16]. The step-by-step molecular mechanisms of LL-37’s antibacterial activity are still unclear. Nevertheless, previous studies indicate that LL-37 operates a multistep mechanism based on three main mechanical models: the carpet model, the barrel stave model, and the toroidal model [15]. The mechanistic pathway of the tetrameric form involves the breakdown of transmembrane potential, which is driven by peptide integration to the membrane as conducting channels [41]. Part of LL-37 (residues 17–29) forms extensive, ribbon-like, thermostable fibrils in solution as a supramolecular structure, colocalizing with bacterial cells. Structure-guided mutagenesis analyses emphasized the role of self-assembly in antibacterial activity [42]. The above findings show that the differences in LL-37 conformations can lead to different types of mechanisms. In future work, molecular dynamics simulation will be used to investigate each conformation, and the mechanisms that they employ, up to the microsecond scale, to observe LL-37’s molecular mechanism involved in damaging the cell membrane.

## 4. Materials and Methods

### 4.1. LL-37 and Membrane System Construction

The LL-37 (LLGDFFRKSKEKIGKEFKRIVQRIKDFLRNLVPRTES) structure was retrieved from the Protein Data Bank (PDB ID: 2K6O) based on NMR spectroscopy analysis [33]. Bacterial cell membrane mimics were represented by 1-palmitoyl-2-oleoyl phosphatidylethanolamine (POPE) and 1-palmitoyl-2-oleoyl phosphatidylglycerol (POPG), the primary lipid components of the inner membrane of bacteria, which were determined to represent around 80% and 20% in experimental research [17]. The 2:1 POPE:POPG lipid composition, which represented the fundamental character of most Gram-negative bacterial membranes, was used as the membrane model [18,43]. We also used 1-palmitoyl-2-oleoyl-sn-glycero-3-phosphocholine (POPC) as a model for the human cell membrane. PACKMOL in Amber20 was used to construct a lipid bilayer, with the size and composition of the systems generated automatically as a model box with 150–170 lipid molecules in each layer (Figure 7). The neutralizing ion, K+ Cl- 0.15 M, as well as explicit solvents in the form of TIP3P water molecules, were also employed [44,45].

### 4.2. Molecular Dynamics Simulations

The 600-ns-long simulations were conducted using the ff14SB, gaff2, and lipid17 forcefields in Amber20 for the POPC and POPE:POPG membrane systems, with various minimization stages performed to obtain the lowest energy. For each system, one LL-37 peptide was placed at 10 Å from the membrane surface, and 150–170 lipid molecules in the upper and lower leaflet of the membrane were modeled. The Langevin thermostat was used for the initial heating in the NVT ensemble, followed by an anisotropic Barendsen weak-coupling barostat, applied at up to 300 K, to simulate a room temperature of approximately 27 °C. Position restrained protein in the equilibration simulations were run in the NPT ensemble for 500 ps and duplicated until they reached 3 ns in duration, followed by the final 500 ps equilibration stage, with all position restraints removed [46]. A timestep of 2 fs and the SHAKE algorithm were used. The particle mesh Ewald method (PME) was used to handle long-range electrostatics, together with the Amber forcefield for protein, lipids, water, and ions. Visual molecular dynamics (VMD) was used to visualize the results of the MD simulation to study the interaction of LL-37 with the membrane surface [47].

### 4.3. Data Analysis

The binding free energy of LL37–POPE:POPG and POPC membranes was calculated using MMPBSA.py in AmberTools21 [48]. An interval of 60 ns and a salt concentration of 150 mM were applied in the calculations. The helicity (%) of LL-37 along the trajectories was analyzed using the Dictionary of Secondary Structure of Proteins (DSSP) method in AmberTools21. Alterations in the POPC and POPE:POPG membranes were analyzed by performing electron density mapping. Distance and SASA analysis were also performed [49]. Data visualization was conducted using BIOVIA Discovery Studio Visualizer, Rstudio, and JupyterLab.

## 5. Conclusions

Atomic-level observation presented the residual interactions of LL-37 and bacterial cell membranes. It showed leucine–leucine as the leading residues that achieve the first contact with the membrane surface. Then, the cationic peptide LL-37 started to attach to the membrane surface by developing emerging salt bridges between positively charged amino acids, Lys–Arg, and the more exposed headgroup of POPE:POPG, which consists of phosphate and hydroxyl groups. Residues 18–29 were observed to be the core region of LL-37, as they actively interact with the POPE:POPG membrane while retaining its helical properties. Interestingly, in terms of selectivity, LL-37 was shown to evade POPC due to the less electronegative choline group as the head group of the membrane. LL-37 in the POPE:POPG membrane presents lower SASA as it is less accessible because of the attachment towards the membrane surface. Our simulation suggests that the core region of LL-37 should be preserved to retain its antimicrobial activity. This study is helpful for understanding the properties and dynamics of LL-37 and to provide a basis for further LL-37 optimization as a biopharmaceutical product.

## Figures and Tables

**Figure 1 ijms-23-13413-f001:**
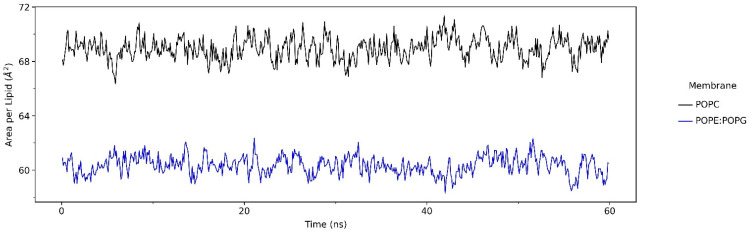
The area per lipid analysis of LL-37 in POPC and POPE:POPG membrane models during 600 ns molecular dynamics simulation. The data were calculated using the NAMD energy module from VMD.

**Figure 2 ijms-23-13413-f002:**
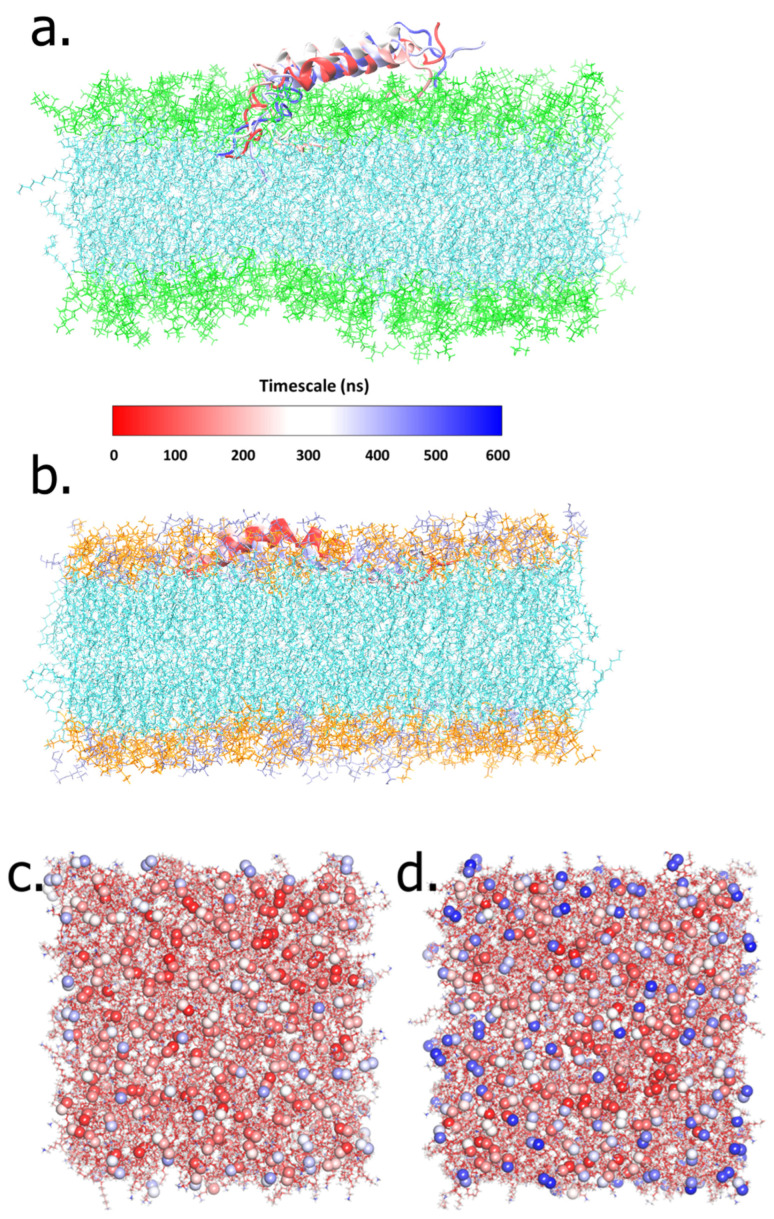
The overall interactions of LL-37 with the POPC (green) (**a**) and POPE:POPG (orange and purple) membrane (**b**) were presented in 150 ns intervals during 600 ns simulations. The exposure of the negatively charged phosphate group visualized by a ball model in (**c**) POPC and (**d**) POPE:POPG bilayer models. The gradual exposure of the water solvent from non-accessible to accessible is visualized by the red- to blue-colored balls.

**Figure 3 ijms-23-13413-f003:**
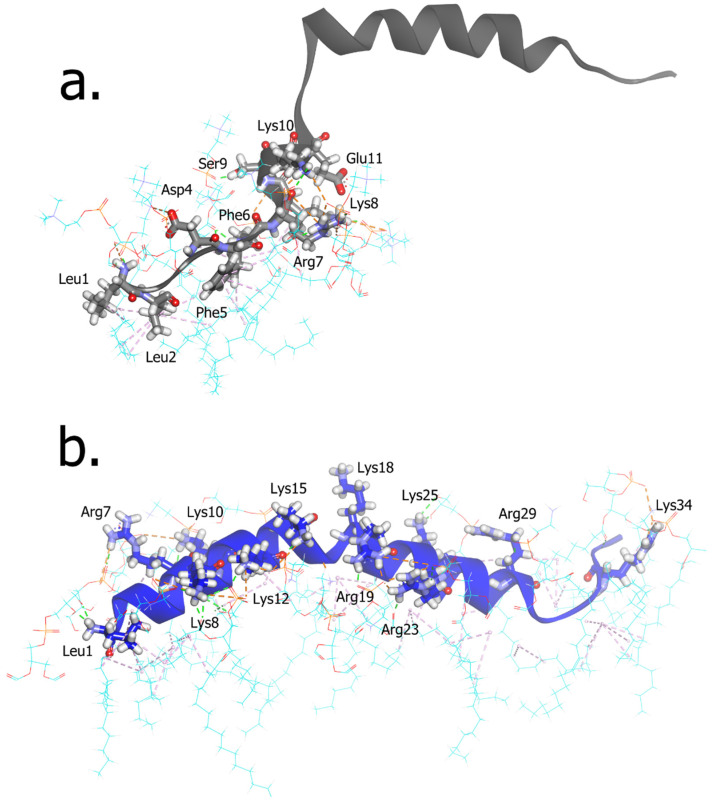
The residual interactions of the N-terminal of LL-37 with POPC (**a**) and the basic amino acids of LL-37 with POPE:POPG membrane (**b**). The structure of LL-37 is shown as black and blue cartoon model; N atoms: blue, O atoms: red, P atoms: orange, tails of phospholipids: cyan.

**Figure 4 ijms-23-13413-f004:**
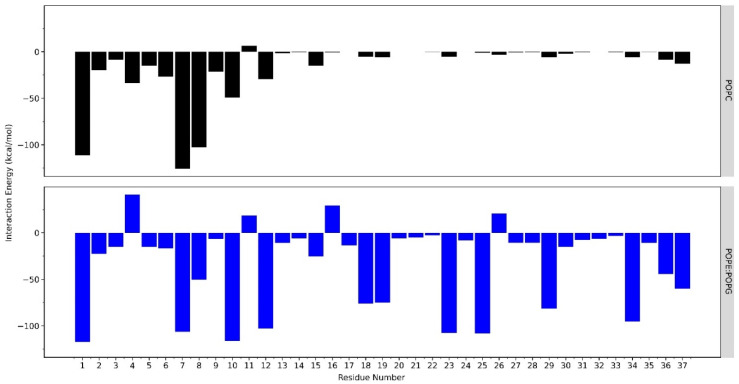
The average interaction energy of each residue of LL-37 during the 600 ns simulation with the POPC and POPE:POPG membrane systems. The data were calculated using the NAMD energy from VMD.

**Figure 5 ijms-23-13413-f005:**
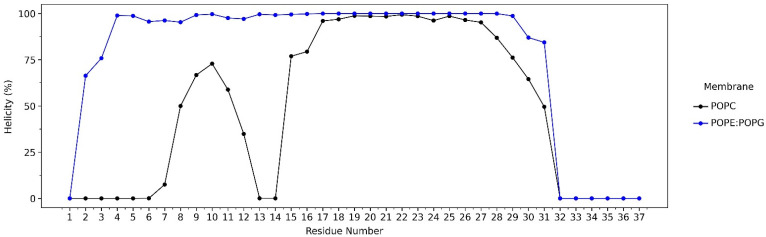
Secondary structure analysis of LL-37 during the 600 ns simulation. In general, LL-37 in the POPE:POPG system retains its helical conformation. The helicity percentage was calculated as the sum of α-helix and 3–10 helix occurrences.

**Figure 6 ijms-23-13413-f006:**
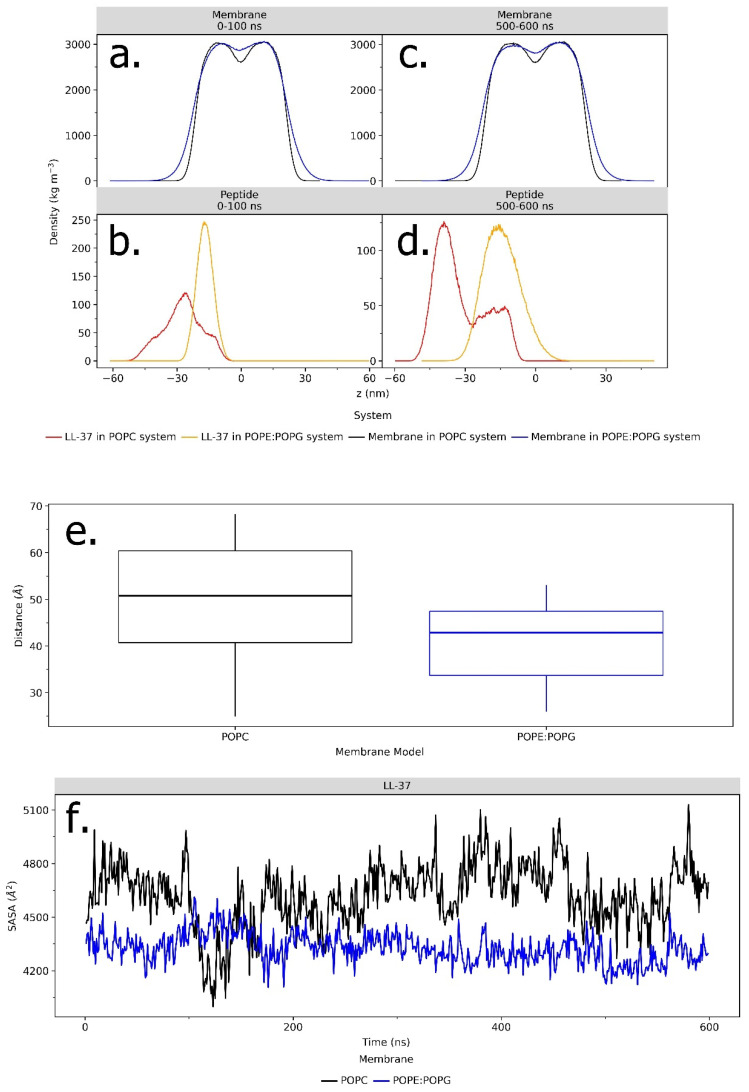
Electron density profiles of LL-37 with the POPC (black) and POPE:POPG (blue) membranes at the initial position and after 600 ns of simulation (**a**–**d**). The average distance (Å) between LL-37 and the center of the POPG and POPC bilayer membranes calculated using cpptraj program in Amber20 (**e**). The SASA (Å2) fluctuation of LL-37 in POPC and POPE:POPG membrane systems during the 600 ns simulation (**f**). The data were calculated using NAMD analysis from VMD.

**Figure 7 ijms-23-13413-f007:**
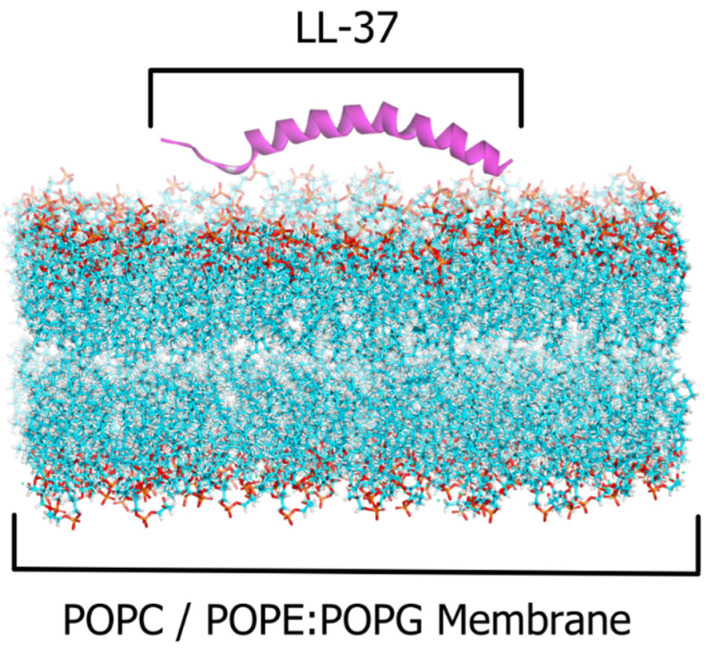
The LL-37–POPC and POPE:POPG membrane simulation system as an AMP–bacterial cell membrane interaction model. The peptide is displayed with purple cartoon models, and the membrane is displayed with solvent models.

**Table 1 ijms-23-13413-t001:** The average binding free energy of LL-37 towards POPC and POPE:POPG calculated by MM/PBSA method in AmberTools21.

Energy Calculations	POPE (kcal/mol)	POPE:POPG (kcal/mol)
E _Van der Waals_	−95.4395	−242.2387
E _Electrostatic_	−627.1922	−6199.7519
E _PB_	641.2761	6208.2515
E _Polar_	−86.3656	−199.5350
ΔG _Gas_	−722.6318	−6441.9906
ΔG _Solv_	703.3229	6356.4950
ΔG _Total_	−19.3089	−85.4955

## Data Availability

The authors declare that all data supporting the findings of this study have been uploaded as part of the electronic Appendix A in the Dryad Repository: https://doi.org/10.5061/dryad.69p8cz93b (accessed on 26 July 2022).

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
