# Peer review of "Residual Interactions of LL-37 with POPC and POPE:POPG Bilayer Model Studied by All-Atom Molecular Dynamics Simulation"

_ijms, 2022, doi:10.3390/ijms232113413_

Round 1
Reviewer 1 Report
The paper presents insufficient data and lacks originality in order to be eligible for publication into IJMS. Below are some comments that might help the authors to improve their manuscript.
The authors provide insufficient/inadequate support of their claim that "In this simulation, residues 18–29 were observed to be the crucial region 197 of LL-37, as they actively interact with the POPG membrane while retaining its helical 198 properties (Figures 2b and 3b)."
Below are pointed some of my main issues regarding the inappropriatness for publication of this manuscript (found in the order they appear in the paper). These are related to (i) faulty approach/lack of proper motivation of using it (see 1, 2), (ii) inappropriate/incomplete analysis of data (see 3, 7), (iii) lack of originality (see 4), (iv) incorrect/incomplete interpretation of analysis of data (see 5, 6).
1) Authors should add reference(s) for the previous studies they refer to in the abstract regarding the MD simulations showing penetration of bilayer membrane, as well as the KR-12 region as being the active part of the LL-37 peptide (both in the abstract and in the text, e.g. sec. 2.1).
2) Exclusively PG membranes are inaccurate descriptions for bacterial membranes. While the AMPs might sort them out of the bilayer, this is working fine, as long as one has the rest of the components at the PBS. Otherwise, artificially high electrostatic repulsive interactions between lipids, and/or bilayer surface tension might occur, which is not acceptable. To eliminate such issues, a mixture of at least 2 lipids, one of which PG in minority (e.g. PE:PG 2:1, to mimic E.coli) is a minimal requirement for a bacterial membrane model (e.g. ref. [G. Shahane, et al. J. Mol.Model. 25 (2019), 76]).
3) An appropriate minimal analysis of the simulated bilayers' behaviour (e.g. electron density, area per lipid, chain order analyses) should be added and compared to more accurate bacterial membrane models (e.g. ref. [T.J. Piggot, D.A. Holdbrook and S. Khalid, J.Phys.Chem. B 115,45 (2011), 13381-13388]) to check reproducibility of the reported results.
4) It seems to me that the authors are repeating the protocol for studying the LL-37 interaction with PC/PG membranes as used in ref. [22].
5) On the identification of the important residues from the energetic point of view, the authors take into consideration only negative interaction energies, and left out residues 4, 16, 26 and even 37, which oppose to the interaction with the PG membrane. Why is that? Moreover, Lys12 has almost zero interaction with the PG, yet it'is considered important!
6) From the analysis at (5) one should conclude that from the KR-12 sequence, only residues 18, 19, 23, 25 and 29 are important for interaction with PG, while residues 16 and 26 are against it. The authors make no reference to the KR-12 residues' importance. Why is that?
7) Finally, the authors do not take into account the entropy impact on each of the peptides. This is of outmost importance to all the peptides, and free energies should be calculated as well, not just the interaction energies.
8) Also, in the discussion section the authors talk/analyze some aspects that are not approached at all in this paper. For example, they talk a lot about KR-12 peptide, yet not even a single simulation of this has been done. If the authors would compute the peptides' adhesion free energies of LL-37 and KR-12 (and compare them, as well as with the interaction energy analyses), than they could comment on this and on KR-12's antimicrobial activity, and even compare it to experimental MIC/MBC values etc.
Typos/minor issues:
1) 2.1. line 6, "... the 18 - 9 residues, known as part of the KR-12 region ..." should be "... the 18 - 29 residues ..."
Author Response
Dear Prof.,
First of all, I would like to appreciate your constructive review of our manuscript. We have repeated the simulation as you suggested, to use POPE:POPG (2:1) model to represent the bacterial membrane better than the previous model. I'm sorry it takes a quite long time for simulating the new one due to our limited computing facility. In the new system, we observed that the ethanolamine group of POPE formed intramolecular interaction with the phosphate group, thus exposing the negative charge on the surface and attracting the LL-37. We found this interesting because only a depiction of a 2D structure won't be able to discover the intramolecular forces that could explain the specificity of LL-37. Kindly please find our response to your review attached. Thank you very much.

Reviewer 2 Report
In this paper, the authors have examined the effect of peptide LL-37 on POPG and POPC membrane systems using Amber20 forcefield molecular dynamics simulations. They conclude that the peptide insertion starts through interaction between positive charge residues Lysine and Arginine and the negative charge of the phospholipid head group. The core region 18-29 of the LL-37, which is helical, is critical for insertion inside the bilayer.
The paper is publishable, but the authors need to address these before publication:
1) Some important publications should be cited in the introduction, such as Biomacromolecules 2013, 14, 10, 3759–3768, Langmuir 2019, 35, 24, 8167–8173
2) The figure number starts from 2. Change it to 1.
3) In Figures 2 and 3, the background lipid bilayer is not visible. Change that.
4) In figure 8, why is there a spike in the sky blue graph? Is it because of the periodic boundary condition?
5) Is there any evidence of pore formation due to the insertion of peptides in the lipid bilayer?
Author Response
Dear Prof.,
First of all, I would like to appreciate your constructive review of our manuscript. We have repeated the simulation as the other Reviewer suggested, to use POPE:POPG (2:1) model to represent the bacterial membrane better than the previous model. I'm sorry it takes a quite long time for simulating the new one due to our limited computing facility. In the new system, we observed that the ethanolamine group formed intramolecular interaction with the phosphate group, thus exposing the negative charge on the surface and attracting the LL-37. Kindly please find our response to your review attached. Thank you very much.

Round 2
Reviewer 1 Report
1) I STRONGLY SUGGEST authors to compare LL-37 with KR-12 and show BASED ON THEIR OWN RESULTS that this is the most important interacting part of the LL-37 AMP.
2) REMOVE COMPLETELY the free energy calculations discussions from the paper or PERFORM THEIR OWN FREE ENERGY calculations to use that in support of their current results.
3) MAJOR REWRITING OF THE ENTIRE DISCUSSION SECTION is required.
Further comments/details on this below ...........
Minor corrections:
1) Last paragraph of Introduction: should correct to "... , the residual interactions that facilitate how it works are yet to be explained."
2) Figure 1 (Y axis - should remove /lipid (typo) at the end)
3) The labels from a to d are missing in Figure 7.
4) Change label on X-axis of Figure 8 from "Membrane Category" to "Membrane Model"
5) Remove typos from the manuscript, e.g. Second paragraph in the Discussion section, correct typo "Leusin" to "Leucine"
6) Add references to the supplementary material into the text of the main manuscript.
Suggestions to the authors:
1) In Fig.2a and Fig.2b they should emphasize graphically (representation or color) the difference between the two bilayers, i.e. PC (CH3)3 vs PE (H3) headgroups. This would enlighten the reader on how the two headgroups arrange themselves ... (see also literature)
2) Figures 7, 8 and 9 could be combined into one.
3) The authors talk about the "... calculation for LL-37 and POPE:POPG membrane interaction showed negative free energy ...", yet no such calculation was performed. Discussions on this topic should be removed, or free energy calculations should be added, as described below (at major corrections).
4) The Discussion section shows the importance of the KR-12 region in the antimicrobial activity of LL-37, yet no studies have been made with this specific part of the peptide. The authors should have done a simulation only with that and compare the two results in order to talk about this !!! This is followed by a talk about how some unknown peptide MMGP1 is penetrating cells, with no connection to the peptide LL-37 ... no comparisons of any kind! Not acceptable to add this in the paper, if no point/connection to LL-37 is made.
5) Further on, the talk about LL-37 activity is shifted to previous studies, and nothing from the current results is compared to this! (The paragraph "The strategy eliminating hydrophobic ... The alanine and lysine scans also identified 18-29 as key element for activity in the LL-37 structure [38]." Why are not the authors addressing this issue in comparison with their own results?
6) The authors suggest that because of the high mobility of lipid molecules their "number in a certain area constantly changes and cannot be calculated"! That is extremly odd! For example, one can calculate the average number of lipids within the vicinity of the peptide at any point in time! This should give the authors an idea on how the peptide influences their behavior in terms of density by its presence (e.g. as function of insertion depth).
7) Again authors talk about binding affinity, but free energy was not computed. Calculating interaction energy components with NAMD Energy is not the same with the free energy! Entire part regarding binding affinity or free energy should be removed from the manuscript or calculated altogether! (there is available a wide variety of methods)!
Major corrections:
1) Figure 4 reflects the enthalpic interactions of the peptide with the membrane surface. The relevant energies however are the free energies. The authors should compare those values (only the fluctuations of the represented so-called "interaction energy" values are, as expected, very large!!!)
2) These large fluctuations show that one cannot take only bonded and non-bonded interactions into account, but free energy calculations (along one or two reaction coordinates (e.g. LL-37 center of mass the distance to the membrane)) are needed to understand how LL-37 interacts with the model bilayers. (the authors nicely suggest that entropic calculations are missing - however, that does not solve the problem).
3) In spite of the wealth of data from the MD simulations, no discussion around the conformational differences of the peptide on the two membranes is performed (except at the results section). No discussion on the interaction of each residue and comparison to previous results... so the real question is what is the novelty here? The authors do not underline this in the discussion (e.g. focus on Fig. 3, 5, 6, 7-9), but keep on talking about results that are not of their own... many times with unclear correlation or connection to what they are reporting here.
Author Response
Dear Prof.,
We would like to thank you for your constructive criticism of our manuscript. We have revised the manuscript following your suggestions, including the calculation of free energy to replace the NAMD energy. Please see the attachment. Thank you again.
Kind regards,
Yusuf
